# JUMP: Jointly Utilizing Missingness for Prediction on Incomplete Tabular Data

## Abstract

Impute-then-predict is the default for tabular data with missing values, yet optimizing reconstruction of imputation rarely guarantees downstream gains and induces distribution shift when train–test missingness differs. We present JUMP, an end-to-end missingness-aware framework that jointly optimizes imputation and prediction. JUMP re-masks a subset of observed features as reconstruction targets, shares a single encoder between reconstruction and prediction heads, and explicitly injects missingness indicators to fuse pattern cues with raw features. This design transforms imputation from a standalone preprocessing step into a training signal that directly serves the predictive objective, acting as a lightweight regularizer that stabilizes representations under missingness. Extensive experiments on eight benchmarks show that JUMP achieves state-of-the-art performance, consistently outperforming twelve impute-then-predict pipelines, strong tree-based models, and advanced neural architectures across diverse missingness mechanisms and challenging out-of-distribution settings.

## 1 Introduction

Tabular data is one of the most prevalent and valuable modalities across both academic and industrial domains, supporting applications in finance, healthcare, customer analytics, manufacturing, and government statistics (Guo et al., 2021; Chen et al., 2016; Sadar et al., 2023; Abdou & Pointon, 2011). However, missing values are ubiquitous due to factors such as collection costs, privacy regulations, sensor malfunctions, and manual entry errors. If not properly addressed, missingness can lead to reduced sample size, biased estimation, and substantial degradation in the stability and generalization of downstream predictive models. Thus, effective and robust handling of missing values is fundamental to building trustworthy tabular machine learning systems.

Traditional approaches to missing-value imputation treat it as a preprocessing step prior to modeling. Simple strategies, including mean, median, or mode substitution and classical statistical or machine learning techniques (e.g. regression imputation), are efficient but often fail to capture intricate inter-feature dependencies, thereby introducing bias or distorting data distributions. More sophisticated statistical methods, such as Multiple Imputation by Chained Equations (van Buuren & Groothuis-Oordshoorn, 2011) and MissForest (Stekhoven & Bühlmann, 2011), offer improved correlation modeling. Recently, deep generative models—including Variational Autoencoders (Kingma & Welling, 2019), Generative Adversarial Imputation Nets (Yoon et al., 2018), and diffusion-based imputers—have shown promise, while self-supervised approaches like ReMasker leverage masked reconstruction to further enhance performance.

Despite these advances, the prevailing workflow for tabular modeling with missing values adheres to a two-stage "impute-then-predict" paradigm. In the first stage, an imputer is trained and evaluated primarily on reconstruction fidelity, often measured by metrics like RMSE. In the second stage, a predictive model is built upon the completed data. This paradigm, however, suffers from two fundamental limitations. First, its objectives are misaligned: higher reconstruction accuracy does not guarantee better downstream performance and may even harm it. Our empirical studies confirm this misalignment, showing that methods excelling at RMSE as shown in Figure 1, such as GAIN, MICE, or MissForest, can underperform simpler alternatives on classification and regression tasks. This occurs because excessive focus on pointwise accuracy can obscure discriminative information

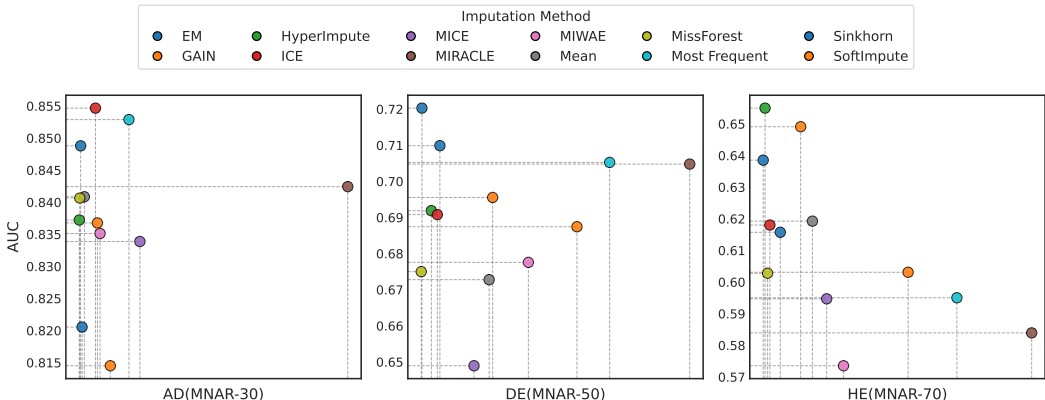

Figure 1: Performance comparison of various imputation methods for downstream prediction tasks based on XGBoost, with varying missing data ratios. The $x$-axis denotes reconstruction error (RMSE on missing entries).

critical for decision boundaries. Second, the decoupling of imputation from prediction prevents the imputer from adapting based on downstream feedback, restricting holistic optimization.

These observations raise a central question: what constitutes good imputation for tabular data? We argue that imputation quality should not be defined solely as approximating unobserved ground truth, but instead as maximizing downstream utility—the ability to improve predictive performance. Addressing this challenge requires breaking the rigid separation between imputation and prediction.

To this end, we propose JUMP, an end-to-end multi-task learning framework that unifies missing-value imputation and prediction. Drawing inspiration from masked autoencoders, JUMP explicitly incorporates missingness indicators and learns adaptive feature–missingness fusion guided directly by the predictive objective. By allowing gradient signals from the prediction task to steer the imputation process, JUMP departs from the sole pursuit of reconstruction fidelity and instead learns to reconstruct information most beneficial to downstream performance. This design overcomes the inherent bottlenecks of two-stage methods and enables consistent performance gains.

Our contributions are threefold:

1. **Rethinking Evaluation:** We provide the first systematic analysis demonstrating the misalignment between reconstruction metrics and downstream task performance in two-stage paradigms, and advocate for a task-utility-driven definition of "good" imputation.

2. **End-to-End Framework:** We introduce JUMP, a novel multi-task architecture that jointly optimizes imputation and prediction, enabling task-aware imputation through shared representations and dual-objective training.

3. **Extensive Validation:** Through comprehensive experiments on public benchmarks under varying missingness mechanisms and rates, we show that JUMP consistently outperforms state-of-the-art two-stage methods, delivering superior predictive accuracy and robustness.

## 2 RELATED WORK

This section reviews prior research on missing-value imputation and tabular prediction, and situates our work at the intersection of these areas to highlight the gap we aim to bridge.

### 2.1 MISSING-VALUE IMPUTATION

In the field of tabular data imputation, existing methods can be categorized into statistical methods, shallow machine learning methods, and deep learning approaches. Statistical methods include mean and mode imputation, which are widely used due to their simplicity and ease of implementation but may introduce bias. Shallow machine learning methods, such as k-Nearest Neighbors

(kNN) imputation, MICE van Buuren & Groothuis-Oudshoorn (2011), and MissForest Stekhoven & Bühlmann (2011), have demonstrated effective performance in filling missing data. Deep learning methods have gained significant attention in recent years. On one hand, neural networks leveraging deep models to uncover causal structures have been applied to data imputation, such as Generative Adversarial Imputation Networks Yoon et al. (2018) and Multiple Imputation with Variational Autoencoders Mattei & Frellsen (2019), which utilize Generative Adversarial Networks and Variational Autoencoders to enhance imputation capabilities.

## 2.2 Tabular Modeling

Traditional approaches remain crucial for supervised and semi-supervised learning on tabular data, with tree-based methods long dominating the field. Tools such as XGBoost (Chen & Guestrin, 2016), CatBoost (Dorogush et al., 2018), and LightGBM (Ke et al., 2017) have achieved widespread success in numerous real-world applications. In recent years, propelled by advances in deep learning—particularly the breakthroughs of Transformers in computer vision and natural language processing—neural network methods for tabular prediction have rapidly emerged. Representative models include TabTransformer (Huang et al., 2020) with attention-centric architectures; TabNet (Arik & Pfister, 2019) with interpretable feature selection and sparse gating; FT-Transformer (Gorishniy et al., 2021) employing bidirectional attention over features and samples; and Transformer variants such as SAINT (Somepalli et al., 2021) that incorporate masked reconstruction or contrastive objectives. These methods leverage self-attention and embeddings to model higher-order feature interactions and, under semi/self-supervised regimes, use masked reconstruction to strengthen representations. However, most deep tabular models still rely on external imputers or simple missingness indicators, with limited focus on unified optimization of imputation and prediction.

## 3 JUMP

In this subsection, we first outline the problem formulation and the underlying missingness mechanisms. We then introduce our novel model **JUMP**. **J**ointly training on value imputation and the primary prediction task creates a unified objective. The core idea is that **U**tilizing the information inherent in data's absence should be guided by the final prediction goal. To achieve this, **M**issingness patterns are explicitly modeled through our proposed Re-Masking mechanism. As a direct result, **P**erformance is enhanced because the model learns to leverage missingness patterns that are truly relevant for the **P**rediction task.

### 3.1 Problem formalization

**Problem Setting** We consider a tabular dataset consisting of $n$ samples and $d$ features. The dataset consists of $n$ samples and $d$ features. For sample $i$, the latent complete feature vector is $x_i = (x_{i1}, \ldots, x_{id}) \in \mathcal{X}_1 \times \cdots \times \mathcal{X}_d$, where each feature space $\mathcal{X}_j$ is either continuous or categorical. Observational access is governed by a missingness mask $m_i = (m_{i1}, \ldots, m_{id}) \in \{0,1\}^d$: $m_{ij} = 1$ indicates that feature $j$ is observed, while $m_{ij} = 0$ indicates missingness (denoted NA). Accordingly, the observed input for sample $i$ is represented as $(x_i^{\text{obs}}, m_i)$, where $x_i^{\text{obs}}$ contains only the entries with $m_{ij} = 1$. Each sample is associated with a supervision signal $y_i$ taking values in $\mathcal{Y}$, which is either $\mathbb{R}$ (regression) or a finite label set (classification). The training set is $\mathcal{D} = \{(x_i^{\text{obs}}, m_i, y_i)\}_{i=1}^n$. Our goal is to learn a predictor $f$ that takes $(x^{\text{obs}}, m)$ as input and predicts $y$ as accurately as possible, i.e., to minimize the expected loss $\mathbb{E}\big[\ell\big(f(x^{\text{obs}}, m), y\big)\big]$ under the data-generating distribution.

**Missingness mechanisms.** Missing entries arise for a variety of reasons. To emulate different scenarios, and following prior work (Yoon et al., 2018; Jarrett et al., 2022), we consider three canonical mechanisms: (1)**MCAR** (missing completely at random): the mask is independent of the data, i.e., $p(m \mid x) = p(m)$ for all $x$ (equivalently, for all $m, x, x'$, $p(m \mid x) = p(m \mid x')$). (2) **MAR** (missing at random): the mask may depend on the observed components of $x$ but not on the unobserved ones; formally, $p(m \mid x) = p(m \mid x_{\text{obs}})$. Hence, if two inputs $x$ and $x'$ share the same observed values, then $p(m \mid x) = p(m \mid x')$. (3) **MNAR** (missing not at random): the mask may also depend on the missing values themselves; this is the case whenever the MCAR and MAR conditions do not hold.

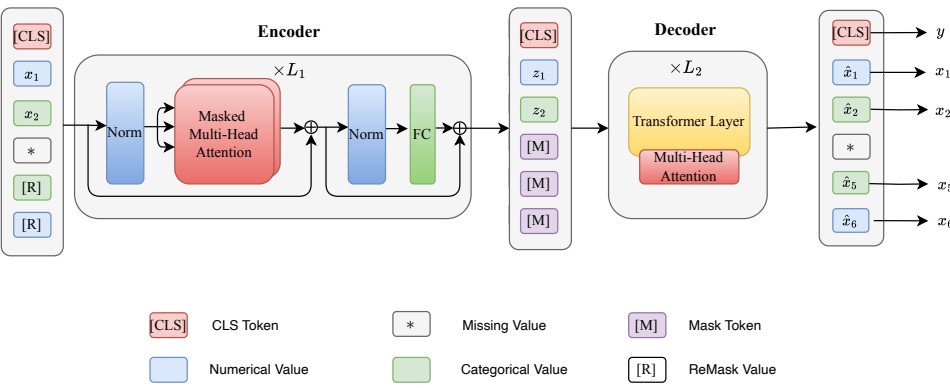

Figure 2: Overall framework of JUMP. During training, for each sample, in addition to the original missing values, we randomly re-mask a subset of observed values. The encoder processes only the remaining visible values to produce representations, which are then padded with learnable mask tokens at the masked positions and passed to the decoder to reconstruct the re-masked values. The [CLS] representation at the decoder output is then used for target prediction.

It is important to note that under the MNAR setting, the missingness distribution is generally not identifiable from observed data alone without imposing additional domain-specific assumptions or structural constraints (Chen et al., 2023).

## 3.2 DESIGN OF MODEL

Inspired by MAE's success on inpainting, we apply a masking mechanism to tabular data with missing values. Because tabular datasets are inherently incomplete, we further re-mask a subset of observed entries to strengthen the learning signal. JUMP adopts an "encoder–decoder + re-masking" framework, augmented with a supervised tabular prediction head, and is trained end-to-end to jointly perform missing-value imputation and target prediction. During training, the re-masking step creates a harder self-supervised objective, encouraging representations that are invariant to missingness patterns. At inference, re-masking is disabled; a single forward pass produces both imputed features and target predictions. The architecture consists of the following modules:

**Re-masking mechanism.** Inspired by MAE He et al. (2021), to construct a more challenging self-supervised learning objective we introduce a *re-masking* mechanism during training, which artificially increases missingness and encourages representations robust to diverse missingness patterns.

Concretely, for each training sample, in addition to its natural missing matrix $m$, we generate a secondary mask $m' \in \{0, 1\}^d$ by uniformly sampling without replacement from the indices of currently observed features. The interaction between $m$ and $m'$ induces three disjoint index sets:

$$I_{\text{missing}} = \{ j \mid m_j = 0 \}, I_{\text{remask}} = \{ j \mid m_j = 1 \wedge m'_j = 0 \}, I_{\text{unmask}} = \{ j \mid m_j = 1 \wedge m'_j = 1 \}.$$

During training, only features in $I_{\text{unmask}}$ are provided to the encoder using their true value embeddings, while all masked positions ($I_{\text{mask}} \cup I_{\text{remask}}$) are initialized with a shared learnable [MASK] token. At inference time, no re-masking is applied (equivalently, $m'$ is all ones). The model leverages all originally observed features, $I_{\text{obs}} \triangleq \{ j \mid m_j = 1 \} = I_{\text{unmask}} \cup I_{\text{remask}}$, to perform imputation and prediction, thereby fully exploiting the available information.

## 3.3 ENCODER

The encoder maps each input value to a vector representation and processes the resulting sequence with Transformer blocks. For numerical features $i$, we use a linear encoding function $e_i^{num} = W_i x + b_i$, where $W_i$ and $b_i$ are learnable parameters. For categorical feature $j$, the embedding is defined as $e_j^{cat} = b_j + E_j^{cat}(x_j^{cat})$, where $E_i^{cat}$ is a learnable embedding table. We also add positional

Figure 3: The proposed masked self-attention mechanism, designed to effectively ignore the impact of missing entries within the attention matrix.

encoding to the embedding of $x$ to make the model memorize $x$'s position in the input (e.g., the $k$-th feature): $\text{pe}(k, 2i) = \sin\left(\frac{k}{10000^{2i/d}}\right)$, where $k$ and $i$ denote the position of $x$ in the input and the embedding dimension index, respectively, and $d$ is the embedding width. After obtaining the feature embeddings and concatenating them into a sequence $\mathbf{E}$, which will be fed to the encoder later in the model architecture. The Transformer computes the query, key, and value matrices—denoted as $\mathbf{Q}$, $\mathbf{K}$, and $\mathbf{V}$ respectively—through linear transformations of the input embedding matrix $\mathbf{E} \in \mathbb{R}^{n \times d_e}$. For each attention head, these projections map the embeddings into a lower-dimensional subspace with dimension $d_h = d/h$, where $h$ is the number of heads. $\mathbf{Q} = \mathbf{E}\mathbf{W}_Q, \mathbf{K} = \mathbf{E}\mathbf{W}_K, \mathbf{V} = \mathbf{E}\mathbf{W}_V$, where the learnable weight matrices are $\mathbf{W}_Q, \mathbf{W}_K, \mathbf{W}_V \in \mathbb{R}^{d_e \times d_h}$, resulting in $\mathbf{Q}, \mathbf{K}, \mathbf{V} \in \mathbb{R}^{n \times d_h}$. The scaled dot-product attention is then computed as:

$$\text{Attention}(\mathbf{Q}, \mathbf{K}, \mathbf{V}) = \underbrace{\text{softmax}\left(\frac{\mathbf{Q}\mathbf{K}^T}{\sqrt{d_h}} + \mathbf{M}\right)\mathbf{V}}_{\text{Attention Weights } \mathbf{A} \in \mathbb{R}^{n \times n}} \tag{1}$$

The matrix $\mathbf{M} \in \mathbb{R}^{n \times n}$ is the attention mask, which adds $-\infty$ to the attention logits corresponding to positions that should be ignored. This operation effectively nullifies their contribution after the softmax function is applied. Instead of applying standard global self-attention, we introduce a Customized Asymmetric Attention Mask specifically engineered for our joint prediction and imputation task.

**Mask Attention In Encoder**  We insert a `[CLS]` token into the encoder for tabular prediction. The input sequence contains four types of tokens: (i) the `[CLS]` token for global aggregation; (ii) *missing* tokens corresponding to originally missing values; (iii) *remask* tokens for entries re-masked during training; and (iv) *unmask* tokens for observed values. Let $I_{\text{unmask}}$, $I_{\text{remask}}$, and $I_{\text{miss}}$ denote the index sets of unmask, remask, and original-missing tokens, respectively. We implement MASK Attention via an attention mask $M$, where disallowed query–key pairs are set to $-\infty$. Under the MASK Attention design, the attention mask follows these rules: (1) The `[CLS]` token has a global view over all non-original-missing entries. Its Query is allowed to attend to itself, unmask tokens, and remask tokens, while positions corresponding to original missing tokens are set to $-\infty$ to prevent leakage from genuinely absent features. (2) Unmask tokens serve as inputs to the reconstruction objective and may only attend to [CLS] and other unmask tokens. Their attention to original missing and remask tokens is set to $-\infty$, disallowing access to invisible or re-masked information. (3) Original missing and remask tokens do not participate in attention during encoding. In the subsequent decoder, they are replaced by a learnable `[MASK]` token to enable imputation and reconstruction. Through this carefully engineered attention mask, we ensure that the `[CLS]` token learns a high-quality global representation for prediction, while the self-supervised reconstruction task proceeds efficiently without any risk of information leakage.

## 3.4 DECODER

The JUMP decoder comprises a stack of Transformer blocks followed by a final MLP layer. Unlike the encoder, the decoder operates on embeddings of both observed and masked values. Following prior work (He et al., 2021), we use a shared, learnable mask token `[MASK]` as the initial embedding for each masked entry ($I_{remask}$ and $I_{missing}$). The decoder adds positional encodings to all value

embeddings (observed and masked), processes them through the Transformer stack, and applies a linear projection to produce scalar predictions.

For supervised prediction, we take the [CLS] representation at the decoder output as the sample-level aggregate and feed it into a task-specific prediction head: an MLP classifier for classification or a regressor for regression, yielding the final output. Crucially, the decoder-end [CLS] has integrated both the originally observed features ($I_{obs}$) and the information inferred for missing parts by the decoder, enabling prediction with the full breadth of tabular information—including signals carried by missingness—which is key to the effectiveness of our model.

### 3.5 JOINT OPTIMIZATION OF RECONSTRUCTION AND PREDICTION

A key contribution of our model is its end-to-end joint training paradigm, which unifies self-supervised missing-value reconstruction with the supervised downstream prediction within a single optimization objective. This design avoids the error compounding inherent in "impute-then-predict" pipelines and enables synergistic learning between the two tasks. We optimize a joint loss:

$$\mathcal{L}_{\text{total}} = \mathcal{L}_{\text{pred}} + \alpha \cdot \mathcal{L}_{\text{recon}}$$

Here, $\mathcal{L}_{\text{pred}}$ is the primary supervised objective, defined as the Cross-Entropy loss for classification or Mean Squared Error (MSE) for regression. The term $\mathcal{L}_{\text{recon}}$ serves as an auxiliary self-supervised objective, computed exclusively on the re-masked set ($I_{\text{remask}}$) to drive the model to learn the data's intrinsic structure. To handle mixed data types, $\mathcal{L}_{\text{recon}}$ is further decomposed into an MSE loss for numerical features ($\mathcal{L}_{\text{recon}}^{\text{num}}$) and a Cross-Entropy loss for categorical ones ($\mathcal{L}_{\text{recon}}^{\text{cat}}$):

$$\mathcal{L}_{\text{recon}}^{\text{num}} = \frac{1}{|I_{\text{remask}}^{\text{num}}|} \sum_{j \in I_{\text{remask}}^{\text{num}}} (\hat{x}_j - x_j)^2 \quad \text{and} \quad \mathcal{L}_{\text{recon}}^{\text{cat}} = \frac{1}{|I_{\text{remask}}^{\text{cat}}|} \sum_{j \in I_{\text{remask}}^{\text{cat}}} \text{CE}(\hat{\mathbf{x}}_j, \mathbf{p}_j)$$

The hyperparameter $\alpha$ balances the reconstruction term, which serves as a strong self-supervised regularizer that promotes more robust representations under the guidance of the primary prediction objective.

## 4 EXPERIMENTS

In this section, we first present the experimental setup and then, to demonstrate the effectiveness of our approach, we investigate the following key questions:

**Q1: On Effective Imputation Strategies.** What is the most effective strategy for handling missing tabular data? Does lower imputation error necessarily lead to better downstream predictive performance?

**Q2: On Comparative Performance Across Tabular Architectures.** Against a range of state-of-the-art tabular model architectures, does our method achieve superior performance on tabular prediction tasks?

**Q3: On Generalization to Unseen Missingness Patterns.** How does our model's performance degrade when faced with test-time missingness rates that differ from those seen during training—a common out-of-distribution scenario? Does it demonstrate superior generalization compared to other methods?

### 4.1 EXPERIMENTAL SETUPS.

**Datasets** We make use of eight well-known tabular datasets: Adult(AD), Default(DE), Shoppers(SP), Beijing(BJ), News(NS), Covtype(CO), Helena(HE) and Jannis(JA). The dataset properties are summarized in Table 1. Following previous works (Muzellec et al., 2020; Zhao et al., 2023), we study three missing mechanisms: MCAR, MAR, and MNAR. In this section, we only report the performance in the MNAR setting, while the results of the other two settings are in Appendix. In the main experiments, we set the missing rate as r = 70%. For each dataset, we generate 5 masks according to the missing mechanism and report the mean and standard deviation of the imputing performance. We using AUC as the main evaluation metric for the classification task and root mean square error (RMSE) for regression.

Table 1: Dataset properties. Notation: "RMSE" ~ root-mean-square error, "Acc." ~ accuracy.

|  | AD | HE | JA | CO | DE | SP | BJ | NS |
|---|---|---|---|---|---|---|---|---|
| #objects | 48842 | 65196 | 83733 | 581012 | 30000 | 12330 | 43824 | 39644 |
| #num. features | 6 | 27 | 54 | 54 | 14 | 10 | 7 | 46 |
| #cat. features | 8 | 0 | 0 | 0 | 10 | 8 | 5 | 2 |
| metric | Acc. | Acc. | Acc. | Acc. | Acc. | Acc. | RMSE | RMSE |
| #classes | 2 | 2 | 4 | 7 | 2 | 2 | – | – |

**Baselines** We organize the baselines into two categories according to whether they natively support missing values, and evaluate all methods on the same train/validation/test split while reporting average ranks across eight datasets. (1) Natively missing-value-aware models: these consume raw features with NaNs and internally handle missingness without any explicit imputation. This category includes the GBDT family—XGBoost (Chen & Guestrin, 2016), LightGBM (Ke et al., 2017), and CatBoost (Dorogush et al., 2018)—as well as our method. We follow each library's recommended practice for categorical features (e.g., native categorical handling or one-hot encoding), perform standardization/encoding only if needed, and adopt identical early stopping and hyperparameter search budgets for fair comparison.

(2) Impute-then-predict methods: we first fit an imputer on the training set and use it to complete the train/validation/test sets by inference, then train downstream tabular predictors on the imputed data. We consider 13 state-of-the-art imputers—HyperImpute (Jarrett et al., 2022), MI-WAE (Mattei & Frellsen, 2019), EM (García-Laencina et al., 2010), GAIN (Yoon et al., 2018), ICE, MICE(van Buuren & Groothuis-Oudshoorn, 2011), MIRACLE (Kyono et al., 2021), MissForest (Stekhoven & Bühlmann, 2011), Mean, Most_Frequent, Sinkhorn (Muzellec et al., 2020), and SoftImpute (Hastie et al., 2015)—combined with representative tabular predictors, including MLP (Hornik et al., 1989), ResNet (He et al., 2015), DCNv2 (Wang et al., 2021), AutoInt (Weiping et al., 2018), MLP-PLR (Gorishniy et al., 2022). Attention-based models such as TabNet (Arik & Pfister, 2019), FT-Transformer (Gorishniy et al., 2021) and TabTransformer (Huang et al., 2020).

## 4.2 RESULTS

**RQ1: Ours vs. Imputation Then Prediction** In tabular learning, missing values are typically handled as part of data preprocessing, and the impute-then-predict paradigm remains commonplace. However, many imputation methods optimize for reconstruction error (e.g., an L2 metric to the original data). Whether lower reconstruction error reliably translates into better downstream predictive performance has not been systematically validated; Moreover, high-capacity neural imputation methods typically incur substantial computational and time costs. To address this core issue, we conduct a comprehensive evaluation of multiple imputation strategies on downstream prediction using FT-Transformer as a unified backbone under the MNAR-70 benchmark; the results are reported in Table 4.2. The figure 4 show that although HyperImpute and EM rank among the top on pure imputation, their impute-then-predict performance sits only around the middle of the 12 methods considered. In contrast, simple mean imputation proves surprisingly robust within the two-stage pipeline and serves as an effective preprocessing baseline. Beyond these two-stage baselines, our method achieves the best performance across all datasets, substantially outperforming competing approaches.

**RQ2.Ours vs. Various Backbones** We conduct a systematic comparison across a suite of modern tabular models spanning multiple architectures: gradient-boosted decision trees (e.g., XGBoost, LightGBM), deep learning models (e.g., ResNet, DCN2, TabNet, FT-Transformer, TabTransformer), and common baselines (e.g., MLP). All models are evaluated under the MNAR-70 missingness setting with a unified preprocessing pipeline: mean imputation for numerical features and mode imputation for categorical features. As shown in Table 4.2, our method achieves the lowest average rank of 1.75 across all datasets, substantially outperforming tree-based models that are strong contenders in tabular learning. Moreover, building on the FT-Transformer backbone, the introduction of the RE-mask mechanism and the joint optimization of observable-value reconstruction lead to marked gains in tabular prediction: an average AUROC improvement of 1.97 percentage points across six classification tasks, and significant RMSE reductions on regression datasets—most notably, over a 32% drop on the beijing dataset.

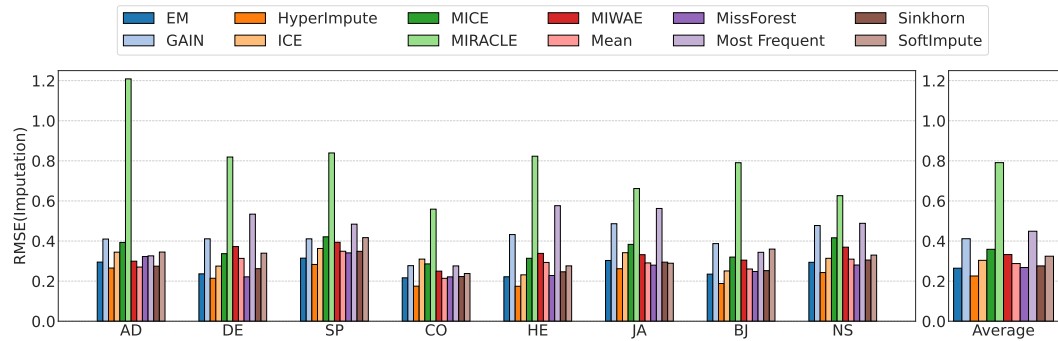

Figure 4: Comparison of imputation methods on eight benchmark datasets under MNAR with a 70% missingness rate. The left panel reports per-dataset performance, and the right panel shows the average across all datasets.

Table 2: The performance of different two-stage prediction methods under the MNAR-70 missingness setting, where all two-stage models use FT-Transformer as their backbone. For classification tasks, we report AUROC (↑ indicates higher is better); for regression tasks, we report RMSE (↓ indicates lower is better). Reported values are averaged over 5 random seeds. For each dataset, the best score is typeset in bold; ranks are assigned by sorting scores from best to worst for each dataset; the "Rank" column reports the average rank across all datasets.

| Model | AD ↑ | DE ↑ | SP ↑ | CO ↑ | HE ↑ | JA ↑ | BJ ↓ | NS ↓ | Avg.Rank |
|---|---|---|---|---|---|---|---|---|---|
| EM | 80.49 | 68.31 | 77.37 | 85.47 | 75.24 | 71.72 | 1.1723 | 0.7373 | 6.43(1.72) |
| GAIN | 75.00 | 65.75 | 77.26 | 85.58 | 75.01 | 71.41 | 1.1874 | 0.7367 | 8.81(2.81) |
| ICE | 77.83 | 68.63 | 77.41 | 85.76 | 75.12 | 72.26 | 1.1901 | 0.7332 | 6.00(2.68) |
| MICE | 75.81 | 67.02 | 75.32 | 85.46 | 75.45 | 71.69 | 1.1815 | 0.7332 | 7.81(2.51) |
| Mean | 81.14 | 68.67 | 77.75 | 85.86 | 75.71 | 72.62 | 1.0558 | 0.7435 | 3.50(2.83) |
| MIWAE | 76.18 | 67.18 | 75.91 | 85.58 | 75.35 | 71.12 | 1.1892 | 0.7468 | 9.00(2.00) |
| MIRACLE | 80.82 | 65.36 | 75.17 | 85.16 | 78.02 | 71.73 | 1.1719 | 0.7474 | 7.62(4.5) |
| MissForest | 80.67 | 69.22 | 77.49 | 85.53 | 75.08 | 72.48 | 1.1819 | 0.7281 | 5.12(3.89) |
| Most_Frequent | 78.9 | 65.87 | 76.75 | 85.77 | 76.71 | 71.41 | 1.2043 | 0.7271 | 6.93(3.86) |
| Sinkhorn | 81.06 | 68.31 | 75.02 | 85.58 | 75.12 | 71.51 | 1.1887 | 0.7382 | 7.68(2.77) |
| SoftImpute | 76.31 | 68.12 | 77.06 | 85.37 | 76.04 | 71.51 | 1.1915 | 0.7369 | 8.06(2.33) |
| JUMP(ours) | **81.62** | **70.48** | **78.78** | **88.41** | **78.63** | **75.99** | **0.7136** | **0.6382** | 1.00(0.00) |

Table 3: Performance of various models on different datasets under MNAR-70 setting. All models use the same preprocessing; numerical features are imputed with the mean, and categorical features with the mode.Reported values are averaged over 10 random seeds. For each dataset, the best score is typeset in bold.

| Model | AD ↑ | DE ↑ | SP ↑ | CO ↑ | HE ↑ | JA ↑ | BJ ↓ | NS ↓ | Avg.Rank |
|---|---|---|---|---|---|---|---|---|---|
| CatBoost | 81.93 | 69.12 | 77.76 | 87.44 | 75.16 | 75.56 | 0.9283 | 0.7223 | 5.68(2.63) |
| LightGBM | **82.54** | 69.56 | 78.46 | 88.08 | 75.52 | 75.84 | 0.9393 | 0.7272 | 5.00(2.51) |
| XGBoost | 82.13 | 69.07 | 77.59 | 88.14 | 76.93 | 76.04 | 0.9286 | 0.7206 | 4.12(2.90) |
| MLP | 78.65 | 63.27 | 78.49 | 86.16 | 75.39 | 72.51 | 1.0132 | 0.7278 | 9.31(3.08) |
| MLP-PLR | 82.09 | 69.34 | 76.94 | 86.61 | 73.86 | 73.89 | 1.0132 | 0.7258 | 5.93(2.54) |
| Resnet | 79.25 | 63.85 | 78.03 | 88.43 | 76.91 | 74.61 | 1.0143 | 0.7276 | 7.12(3.68) |
| DCN2 | 81.13 | 70.06 | 78.65 | 88.32 | 75.67 | 73.57 | 1.0136 | 0.7264 | 6.00(3.17) |
| AutoInt | 81.22 | 69.92 | 77.70 | 86.67 | 76.01 | 74.03 | 1.0133 | 0.7259 | 6.50(1.77) |
| TabNet | 78.46 | 58.80 | 75.61 | 85.49 | 74.83 | 73.11 | 0.9021 | 0.7245 | 10.00(4.56) |
| Saint | 81.04 | 67.40 | 76.52 | 85.83 | 74.98 | 71.60 | 1.0543 | 0.7441 | 11.62(1.06) |
| TabTransformer | 81.60 | 69.12 | 76.87 | 87.98 | 77.16 | 71.12 | 1.0527 | 0.7432 | 8.31(3.73) |
| FT-Transformer | 81.14 | 68.67 | 77.75 | 85.86 | 75.71 | 72.62 | 1.0558 | 0.7435 | 9.62(2.26) |
| ours | 81.62 | **70.48** | **78.78** | **88.41** | **78.63** | **75.99** | **0.7136** | **0.6382** | 1.75(1.38) |

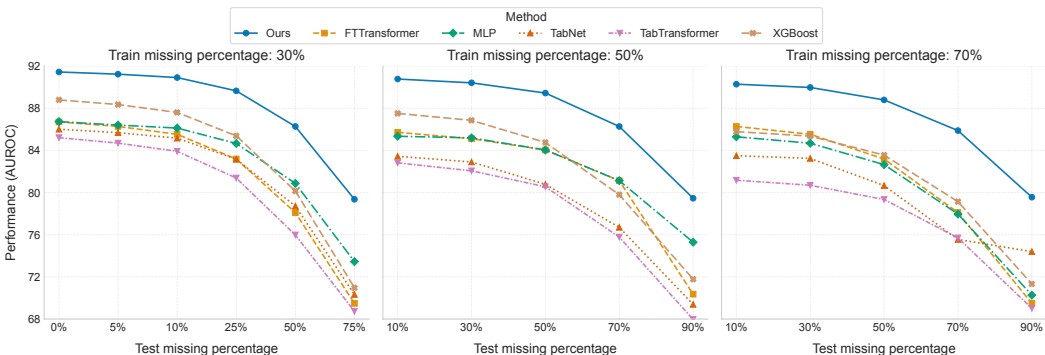

Figure 5: Model performance under varying missingness settings. On the Adult dataset, we evaluate multiple models under mismatched training and test missingness rates. Each subplot fixes the training missingness at 30%, 50%, or 70%, while the x-axis varies the test-time missingness (10%, 30%, 50%, 70%, 90%), enabling a direct comparison of robustness to out-of-distribution (OOD) missingness.

**RQ3. Ours on. OOD test data** To assess generalization under out-of-distribution (OOD) missingness, we conduct a key experiment: models are trained on datasets with missingness rate A (under MAR/MNAR/MCAR) and evaluated on test sets spanning a broader range of missingness rates B; On the Adult dataset, we evaluate test-time missingness rates of 10%, 30%, 50%, 70%, and 90% under MAR/MNAR/MCAR. For each rate, we average performance across the three mechanisms and report the results in Figure 5. while performance degrades as test-time missingness increases, our method consistently leads and exhibits substantially smaller drops. For example, Under 50% and 70% training missingness, our method also shows the smallest degradation even at 90% test missingness. This indicates that our approach learns representations less sensitive to missingness patterns, yielding stronger generalization and predictive performance under unknown test distributions. These gains stem from our core design. Beyond tree models and our approach, most neural methods (TabNet, MLP, FT-Transformer) follow a two-stage pipeline: they pre-impute the test set using training statistics (e.g., means) before prediction. When training and test missingness differ, this step induces significant distribution shift; moreover, imperfect imputations compound this shift, causing substantial performance drops. In contrast, our method natively handles missing data and avoids erroneous preprocessing, delivering superior robustness under OOD missingness patterns.

## 5 CONCLUSION

This work revisits missing-value handling in tabular learning and demonstrates that optimizing imputation in isolation is often misaligned with downstream objectives. We introduced JUMP, an end-to-end framework that jointly optimizes imputation and prediction by re-masking observed features, sharing a single encoder across tasks, and explicitly injecting missingness indicators to fuse pattern cues with raw features. By turning imputation into a training signal that directly serves the predictive objective, JUMP mitigates error propagation, stabilizes representations under varying missingness regimes, and consistently outperforms strong impute-then-predict pipelines, tree-based baselines, and advanced neural architectures across diverse settings. Our analysis and experiments advocate a task-utility-driven view of "good" imputation and show that unified optimization yields tangible gains in accuracy and robustness.

## ETHICS STATEMENT

This study relies solely on publicly available, anonymized datasets from the UCI Machine Learning Repository and does not involve any personally identifiable information or sensitive data. Our work focuses on the foundational technical challenge of handling missing data in tabular learning to improve robustness and trustworthiness. No human-subject interaction was conducted, and no additional IRB approval is required. We used a large language model (GPT) exclusively only for

editorial polishing of the manuscript. All experimental design, implementation, and conclusions were carried out independently by the authors.

## REPRODUCIBILITY STATEMENT

To ensure reproducibility, we document data sources, preprocessing, and training details in the main text and appendix. All datasets are from the UCI Machine Learning Repository; data selection and cleaning procedures, categorical encoding/standardization strategies, train/validation/test temporal splits, and the missing-mask generation mechanisms (parameters and implementations for MCAR/MAR/MNAR) are provided in the appendix (Data and Protocols section). Model architecture and training configurations (encoder/decoder depth, embedding dimensions, optimizer, learning rate, batch size, early stopping criteria), the weighting of the joint loss, hyperparameter search space and budget, as well as evaluation metrics and statistical reporting (mean/standard deviation, average ranks) are clearly specified in the Methods and Experiments sections.

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

## A EXPERIMENTAL DETAILS

**Environment.** All experiments are conducted with 8 GPU V100, Intel(R) Xeon(R) Gold 6240 CPU @ 2.60GHz, and 128GB RAM.

**Parameter Setting.** The default parameter settings for our model, JUMP, are as follows. For global settings, we use the Adam optimizer with an initial learning rate of 1e-3. The learning rate scheduler is set to cosine annealing, and the gradient clipping threshold is 5.0. The model is trained for 600 epochs, with a batch size of 64 and a masking ratio of 0.5. Regarding the model architecture, both the encoder and decoder components of JUMP are based on the Transformer architecture. The encoder consists of 8 Transformer blocks with an embedding width of 64 and 4 heads. The decoder is composed of 4 Transformer blocks, also with an embedding width of 64 and 4 heads.

**Missing Mechanisms.** Missing data can be categorized into three canonical mechanisms based on how the missingness patterns are generated:

1. **Missing Completely at Random (MCAR):** The probability of an entry being missing is independent of any data values, i.e., $p(\mathbf{m}|\mathbf{x}) = p(\mathbf{m})$.

2. **Missing at Random (MAR):** The probability of an entry being missing depends only on the observed values, i.e., $p(\mathbf{m}|\mathbf{x}) = p(\mathbf{m}|\mathbf{x}_{\text{obs}})$.

3. **Missing Not at Random (MNAR):** The probability of an entry being missing may also depend on the unobserved (missing) values themselves. This category encompasses all cases not covered by MCAR or MAR.

To generate realistic missingness patterns for our experiments, we implement the MAR and MNAR mechanisms following the procedure proposed by Zhao et al. (2023):

- **MAR Generation:** We first partition the features into two sets: a fully observed set and a potentially missing set. A logistic regression model is then trained using the fully observed features as input to predict the probability of missingness for each entry in the potentially missing set. The bias term of this logistic model is adjusted via a line search to achieve the desired overall missingness rate.

- **MNAR Generation:** We adopt the "logistic model with MCAR-masked inputs" approach from the reference. Similar to the MAR setup, we divide features into two sets. A logistic model uses the first set of features to predict the missingness probabilities for the second set. Crucially, after these probabilities are determined but before they are used to generate masks, we apply an MCAR mask to the *input* features (the first set). This creates a dependency where the missingness in the second set is influenced by the (now masked) values in the first set, thus satisfying the MNAR condition.

## B IMPLEMENTATION DETAILS

**Baseline of Imputation.** We use 12 state-of-the-art imputation methods:

- HyperImpute: a hybrid imputer that performs iterative imputation with automatic model selection.

- MIWAE: an autoencoder model that fits missing data by optimizing a variational bound.

- EM: an iterative imputer based on expectationmaximization optimization.

- GAIN: a generative adversarial imputation network that trains the discriminator to classify the generator's output in an element-wise manner.

- ICE: an iterative imputer based on regularized linear regression; MICE, an ICE-like, iterative imputer based on Bayesian ridge regression.

- MIRACLE: an iterative imputer that refines the imputation of a baseline by simultaneously modeling the missingness generating mechanism.

- MissForest: an iterative imputer based on random forests.

- Mean and Most_Frequent, which impute missing values using column-wise unconditional mean, median, and the most frequent values, respectively.

- Sinkhorn: an imputer trained through the optimal transport metrics of Sinkhorn divergences.

- SoftImpute, which performs imputation through soft-thresholded singular value decomposition.

**Baseline Implementation.** The setup of our baseline follows the previous work and includes the following methods:

- XGBoost: Implemented based on the XGBoost package. We set the maximum number of estimators in 50, 100, 300 and the max depth in {5, 8, 10}.

- LightGBM: Implemented based on the LightGBM package. We set the maximum number of estimators in {50, 100, 300} and the max depth in {5, 8, 10}.

- MLP: Dense layers with hidden dimensions {256, 256}. Dropout with a rate of 0.1 is used. They are trained with batch size in {16, 32, 64, 128}, learning rate in {5e-5, 1e-4, 1e-3}, and early stopping patience of 5 with 100 maximum epochs.

- TabNet: Use the official implementation with the default recommended parameters. Trained with batch size in {16, 32, 64, 128}, learning rate in {1e-4, 1e-3, 2e-2}, $n_a, n_b$ in {8, 16, 64, 128}, $\gamma$ in {1.3, 1.5, 1.8}, categorical embedding dimension in {1, 8, 16} and early stopping patience of 5 with 100 maximum epochs.

- DCN-v2: The number of cross is 2. The dropout rate for the feedforward component is 0.1. MLP part has two dense layers of dimension {256, 128}. Trained with batch size in {16, 32, 64, 128}, learning rate in {5e-5, 1e-4, 1e-3}, and early stopping patience of 10 in 100 maximum epochs.

- AutoInt: The attention layer number is set to 2. The attention head number is set to 2. MLP part has two dense layers of dimension 256, 128; dropout deactivated; trained with batch size in 16, 32, 64, 128, learning rate in {5e-5, 1e-4, 1e-3}, and early stopping patience of 10 in 100 maximum epochs.

- SAINT: The embedding size is 32 dimensions. 6 transformer layers are used. The number of heads of attention is in {4, 8}. The dropout rate is 0.1 in all attention layers and feed-forward layers. Inside the self-attention layer, the q, k, and v vectors are of dimension 16, and in the intersample attention layer, they are of size 64.

- FT-Transformer: Feed-forward component has 128 dimensions. 2 transformer layers are used. The number of heads of attention is in {2, 4, 8}. The dropout rate is 0.1.

- TransTab: Token embedding has 128 dimensions. 2 transformer layers are used. The number of heads of attention is 8. We train the model on all downstream task data taking batch size 64, learning rate 1e-4, dropout rate 0, and early stopping patience of 10 in 100 maximum epochs. We run the pretraining, transfer learning, and vanilla supervised training methods in the paper, and take the highest score.

## C ADDTIONAL RESULTS

To assess generalization under out-of-distribution (OOD) missingness, we conduct a key experiment in which models are trained at a fixed missingness rate A (under MAR/MNAR/MCAR) and evaluated at test time across a broader range of rates B. On the Adult dataset, we report performance under MAR/MNAR/MCAR for test-time missingness rates of 10%- 90%. We benchmark FT-Transformer, MLP, TabNet, TabTransformer, XGBoost, and our method JUMP on both the Adult and Shopper datasets. Detailed results are presented in the Tables C and C.

Table 4: Performance of various models under different missing settings in ADdult.

| Train-30% | 10% | 30% | 50% | 70% | 90% |
|---|---|---|---|---|---|
| Ours | 91.23 (0.17) | 90.91 (0.17) | 89.65 (0.14) | 86.27 (0.31) | 79.36 (0.29) |
| FTTransformer | 86.27 (0.13) | 85.53 (0.16) | 83.18 (0.23) | 78.10 (0.30) | 69.49 (0.26) |
| MLP | 86.40 (0.18) | 86.12 (0.20) | 84.66 (0.27) | 80.87 (0.25) | 73.45 (0.41) |
| TabNet | 85.68 (0.21) | 85.16 (0.22) | 83.16 (0.25) | 78.72 (0.24) | 70.33 (0.21) |
| XGBoost | 88.35 (0.19) | 87.60 (0.21) | 85.38 (0.25) | 80.15 (0.27) | 70.97 (0.12) |
| **Train-50%** | **10%** | **30%** | **50%** | **70%** | **90%** |
| Ours | 90.77 (0.27) | 90.41 (0.25) | 89.44 (0.26) | 86.27 (0.21) | 79.46 (0.24) |
| FTTransformer | 85.71 (0.17) | 85.11 (0.20) | 82.93 (0.19) | 78.14 (0.27) | 70.36 (0.33) |
| MLP | 85.34 (0.24) | 85.19 (0.18) | 84.04 (0.13) | 81.13 (0.23) | 75.29 (0.22) |
| TabNet | 83.44 (0.36) | 82.90 (0.37) | 80.78 (0.36) | 76.70 (0.30) | 69.39 (0.34) |
| XGBoost | 87.51 (0.21) | 86.85 (0.26) | 84.75 (0.13) | 79.80 (0.24) | 71.78 (0.23) |
| **Train-70%** | **10%** | **30%** | **50%** | **70%** | **90%** |
| Ours | 90.28 (0.17) | 89.97 (0.16) | 88.79 (0.23) | 85.88 (0.27) | 79.56 (0.25) |
| FTTransformer | 86.27 (0.13) | 85.53 (0.16) | 83.18 (0.23) | 78.10 (0.30) | 69.49 (0.26) |
| MLP | 85.29 (0.25) | 84.68 (0.27) | 82.65 (0.27) | 77.96 (0.28) | 70.27 (0.50) |
| TabNet | 83.50 (0.66) | 83.24 (0.59) | 82.54 (0.44) | 80.86 (0.29) | 75.55 (0.40) |
| XGBoost | 85.78 (0.33) | 85.35 (0.30) | 83.55 (0.15) | 79.16 (0.23) | 71.33 (0.39) |

Table 5: Performance of various models under different missing settings in Shoppers.

| Train-30% | 10% | 30% | 50% | 70% | 90% |
|---|---|---|---|---|---|
| Ours | 91.40 (0.37) | 90.66 (0.28) | 88.56 (0.49) | 83.16 (0.47) | 73.15 (0.75) |
| FTTransformer | 89.01 (0.60) | 88.11 (0.56) | 86.25 (0.60) | 81.36 (0.67) | 71.67 (0.48) |
| MLP | 79.10 (0.68) | 78.35 (0.70) | 75.36 (0.68) | 69.13 (0.76) | 61.78 (1.02) |
| TabNet | 88.49 (0.61) | 87.51 (0.61) | 85.27 (0.81) | 80.26 (0.47) | 70.77 (0.86) |
| XGBoost | 89.95 (0.39) | 89.16 (0.35) | 86.98 (0.46) | 82.07 (0.62) | 73.38 (0.54) |
| **Train-50%** | **10%** | **30%** | **50%** | **70%** | **90%** |
| Ours | 90.36 (0.45) | 89.88 (0.53) | 87.74 (0.32) | 82.92 (0.63) | 74.07 (0.69) |
| FTTransformer | 81.11 (7.35) | 80.60 (7.23) | 78.93 (6.85) | 74.58 (5.83) | 67.96 (4.27) |
| MLP | 77.27 (0.94) | 76.63 (0.80) | 74.12 (0.90) | 69.96 (0.86) | 63.57 (0.71) |
| TabNet | 87.42 (0.61) | 87.01 (0.60) | 84.96 (0.76) | 78.86 (0.71) | 70.77 (1.36) |
| XGBoost | 88.75 (0.66) | 88.22 (0.59) | 86.02 (0.77) | 81.58 (0.64) | 73.47 (0.69) |
| **Train-70%** | **10%** | **30%** | **50%** | **70%** | **90%** |
| Ours | 89.08 (0.67) | 88.65 (0.79) | 86.59 (0.62) | 81.39 (0.70) | 73.62 (0.64) |
| FTTransformer | 84.69 (0.84) | 83.88 (0.99) | 81.41 (0.87) | 76.04 (0.92) | 69.08 (0.62) |
| MLP | 73.28 (0.40) | 72.46 (0.62) | 69.91 (0.46) | 70.09 (0.38) | 66.19 (0.60) |
| TabNet | 86.31 (0.67) | 85.99 (0.69) | 83.93 (0.72) | 79.19 (0.48) | 72.09 (0.33) |
| XGBoost | 87.29 (0.61) | 87.02 (0.48) | 84.81 (0.44) | 81.46 (0.32) | 73.72 (0.35) |

