# OpenReview forum: "JUMP: JOINTLY UTILIZING MISSINGNESS FOR PREDICTION ON INCOMPLETE TABULAR DATA"
_ICLR.cc/2026/Conference — ICLR 2026 Conference Withdrawn Submission_

### Official Review · Reviewer_4MqN · 2025-10-29

**Soundness:** 2
**Presentation:** 3
**Contribution:** 2
**Rating:** 4
**Confidence:** 4

**Summary:**

This paper introduces JUMP, an end-to-end framework that jointly optimizes imputation accuracy and downstream task performance. The authors first empirically demonstrate that higher imputation accuracy does not necessarily lead to better downstream task performance and can even degrade it. To resolve this issue, the authors propose a unified model that performs both label prediction and imputation simultaneously, and is trained to jointly optimize imputation accuracy and downstream task performance. JUMP follows prior imputation methods to adopt a remasking mechanism to generate imputation signals. Experimental results on 8 datasets under three missingness mechanisms and various missingness ratios demonstrate that JUMP achieves better downstream task performance compared to existing methods. Moreover, the authors show that JUMP remains robust when tested on unseen missingness ratios.

**Strengths:**

**(S1)** The paper is clearly written and well structured. The design choice of jointly optimizing imputation accuracy and downstream task performance is empirically motivated.

**(S2)** The proposed framework is designed to simultaneously optimize imputation accuracy and downstream task performance. Thus, the imputed value can be more helpful for improving downstream task performance.

**(S3)** The effectiveness of JUMP is empirically validated on eight datasets under three missingness mechanisms and various missingness ratios. The authors also demonstrate that JUMP, when trained on datasets with fixed missingness ratios, can generalize to unseen missingness ratios during testing.

**Weaknesses:**

**(W1)** The technical novelty of the proposed method appears limited. The remasking mechanism used during imputation training has been explored in prior studies [1]. The idea of jointly optimizing imputation performance together with downstream task performance has also been investigated in earlier works.

**(W2)** The experimental settings raise several concerns. First, although JUMP is presented as a framework that unifies imputation and prediction, the paper does not include any empirical comparison with existing imputation methods in terms of imputation accuracy (e.g., MSE). Moreover, in RQ.3, the authors compare JUMP with several tabular models on downstream task performance. However, for these baseline tabular models, missing values are imputed using simple techniques such as mean imputation, whereas JUMP might imputes missing values through its proposed pipeline. To ensure a fair comparison, the authors should consider using more advanced imputation methods (e.g., HyperImpute) for the baseline tabular models

[1] Du, Tianyu, Luca Melis, and Ting Wang. "ReMasker: Imputing Tabular Data with Masked Autoencoding." The Twelfth International Conference on Learning Representations.

**Questions:**

**Q1**. Would it be better to include tick marks on the x-axis in Figure 1, for better clarity?

**Missing reference**:

Du, Tianyu, Luca Melis, and Ting Wang. "ReMasker: Imputing Tabular Data with Masked Autoencoding." The Twelfth International Conference on Learning Representations.

**Incorrect referencing at line 697**.

---

### Official Review · Reviewer_T2ds · 2025-10-29

**Soundness:** 2
**Presentation:** 3
**Contribution:** 2
**Rating:** 2
**Confidence:** 4

**Summary:**

This paper studies how to better conduct downstream tasks, such as classification and regression, for tabular data with missing values. It proposes JUMP, which introduces additional masking of the data during training, and leverages a [CLS] token to attend on observed, missing, and re-masked data using a transformer architecture, such that downstream tasks can be completed in an end-to-end manner. Experiments are done to validate the effectiveness of the design.

**Strengths:**

**S1.** This paper is well written and easy to follow.

**S2.** It is important to have good performance in downstream tasks for tabular dataset with missing values.

**Weaknesses:**

**W1.** The contribution is limited. Re-Masking for missing value imputation has been studied in [1], which also uses an end-to-end pipeline with attention mechanism for downstream tasks.

**W2.** It is mentioned in Line 50 that "higher reconstruction accuracy does not guarantee better downstream performance and may even harm it." Any intuition for that? Also, why there would be a distribution shift "when train-test missingness differs" (Line 13, Line 462)?

**W3.** In Figure 1, why do we use different datasets with various missingness settings? In addition, methods such as EM and HyperImpute seem to perform pretty well with small reconstruction errors and high accuracies in prediction. It is better to include JUMP in this plot for a fair and clear comparison.

**W4.** Is there an ablation study to show the model performance if we drop the [CLS] token in the design? [CLS] token is effective in classification tasks. By dropping the token, we can see the true performance gain achieved by the remasking scheme applied here.

**W5.** About the experiments:

	1. What is the train/validataion/test split?

	2. Usually in missing value imputation, 30% missingness under the MCAR setting is studied as default. More experiments on this may be needed.

	3. From Figure 4, it seems like HyperImpute is the best one? In Table 2, there is no result reported for HyperImpute as well. What is the reason? In addition, the improvement of JUMP from EM does not seem to be significant.

	4. In Figure 5, which missingness pattern is used exactly?

	5. In Line 697, what are Tables C and C? Why are the experiments only conducted on two datasets? There are only two tables and three missingness settings, MAR, MNAR, and MCAR. How do they correspond to each other?

[1] LSM-2: Learning from Incomplete Wearable Sensor Data

**Questions:**

None.

---

### Official Review · Reviewer_Nrma · 2025-10-30

**Soundness:** 2
**Presentation:** 2
**Contribution:** 1
**Rating:** 2
**Confidence:** 5

**Summary:**

This paper proposes JUMP, a framework for handling missing values in tabular data that jointly optimizes imputation and downstream prediction tasks. They propose re-masking a subset of observed features during training and share an encoder between reconstruction and prediction heads. The authors argue that traditional impute-then-predict paradigms are misaligned since reconstruction accuracy does not guarantee downstream performance.

**Strengths:**

1. The paper provides evidence that reconstruction metrics do not correlate with downstream task performance, motivating the need for joint optimization.

2. The asymmetric attention mechanism is an interesting alternative to sequence truncation thats used by ReMasker.

**Weaknesses:**

1. The paper fails to cite ReMasker [1], which is extremely concerning given striking similarities:
    - Both use re-masking of observed values with virtually identical notation (I_missing, I_remask, I_unmask)
    - Transformer encoder-decoder with mask tokens
    - Masked autoencoding for tabular data imputation

2. If we consider ReMasker as prior work (which we must), the core contribution reduces to adding L_pred to the loss function, which is relatively trivial regularizer for task specific modification.

3. The paper also fails to benchmark aginst ReMasker or any other other latest imputation approaches [1,2,3]. All of these can be amended to include L_pred. So, its unclear if this work makes any meaningful contribution to the field.

### References
[1] ReMasker: Imputing Tabular Data with Masked Autoencoding (ICLR 2024)

[2] DiffPuter: Empowering Diffusion Models for Missing Data Imputation (ICLR 2025)

[3] CACTI: Leveraging Copy Masking and Contextual Information to Improve Tabular Data Imputation (ICML 2025)

**Questions:**

n/a

**Details Of Ethics Concerns:**

Seems too similar to ReMasker without any acknowledgment of its existence.

---

### Official Review · Reviewer_FpGJ · 2025-10-31

**Soundness:** 3
**Presentation:** 3
**Contribution:** 3
**Rating:** 8
**Confidence:** 3

**Summary:**

This is the a solid ICLR submission. The authors clearly justify the main problem they aim to address: in tabular datasets with missing values, imputation should be done jointly with prediction to avoid misalignment between the two. They highlight how impute-then-predict methodologies show a clear misalignment between reconstruction error and prediction error. They propose a strategy inspired by MAE inpainting—training a ViT by masking parts of the input, using a “data augmentation” approach that hides different input values within an attention-based model. The paper is well written, well motivated, and presents realistic results, both against methods that handle missing values natively and against advanced imputation-based approaches.

**Strengths:**

The paper clearly introduces and motivates the main problem and its value proposition — namely,
that traditional Impute–Then–Predict techniques suffer from a misalignment between the objectives
of imputation and prediction. Building upon the Masked Autoencoder (MAE) inpainting approach,
this work effectively extends that methodology to tabular data with inherent missingness, leading
to improved downstream performance.

The paper is well written and easy to follow, guiding the reader through its motivation, methodology,
and experiments with clear logic and structure. The integration of transformers with the proposed
re-masking mechanism is well explained, and the joint loss function clearly illustrates how the model
simultaneously addresses both prediction and imputation objectives.
The selection of baseline models is well justified and covers both imputation pipelines and end-to-
end prediction models. Tables 2 and 3 clearly demonstrate that JUMP achieves strong performance
across both two-stage impute–then–predict pipelines and models trained directly on incomplete
data. The main text provides sufficient detail to understand the design choices, contributions, and
empirical value of the approach without requiring heavy reference to the appendix.

**Weaknesses:**

While the results are consistent across datasets, the absolute performance gains are relatively modest, though this makes them more credible and realistic rather than overstated. However, it remains
unclear how the \alpha parameter in the joint loss was tuned, as no ablation or sensitivity analysis is
provided.

No comparison of computational efficiency or runtime overhead is included, which limits assessment
of the practical trade-offs between accuracy and computational cost.

**Questions:**

1. Please correct the table references in Section 4.2, where both are referred to as “Table 4.2” instead of Tables 2 and 3.

2. In Figure 1, GAIN and SoftImpute are both depicted in orange, making them indistinguishable — please use distinct colors.

3. The reference to MAE (He et al., 2021) should appear when the concept is first introduced (around line 188), rather than later at line 197.

4. An analysis of the $\alpha$ parameter in the joint loss would be valuable to understand the sensitivity of the model to this weighting factor and its influence on the balance between reconstruction and prediction.

5. What is the computational overhead of JUMP during both training and inference compared to standard \emph{impute–then–predict} pipelines?

6. Line 417: There is a missing space between “the mode.Reported”.

7. Similarities and differences with "Knockout" can be spelled out. Nguyen, M., Karaman, B.K., Kim, H., Wang, A.Q., Liu, F. and Sabuncu, M.R., 2024. "Knockout: A simple way to handle missing inputs." TMLR

---

### Note · Authors · 2026-01-13

I have read and agree with the venue's withdrawal policy on behalf of myself and my co-authors.